# Perturbation Theory for the Information Bottleneck

**Vudtiwat Ngampruetikorn,**[*]   **David J. Schwab**
Initiative for the Theoretical Sciences, The Graduate Center, CUNY
[*]vngampruetikorn@gc.cuny.edu

## Abstract

Extracting relevant information from data is crucial for all forms of learning. The information bottleneck (IB) method formalizes this, offering a mathematically precise and conceptually appealing framework for understanding learning phenomena. However the nonlinearity of the IB problem makes it computationally expensive and analytically intractable in general. Here we derive a perturbation theory for the IB method and report the first complete characterization of the learning onset—the limit of maximum relevant information per bit extracted from data. We test our results on synthetic probability distributions, finding good agreement with the exact numerical solution near the onset of learning. We explore the difference and subtleties in our derivation and previous attempts at deriving a perturbation theory for the learning onset and attribute the discrepancy to a flawed assumption. Our work also provides a fresh perspective on the intimate relationship between the IB method and the strong data processing inequality.

## 1   Information Bottleneck

Extracting relevant information from data is crucial for all forms of learning. Animals are very adept at isolating biologically useful information from complicated real-world sensory stimuli: for example, we instinctively ignore pixel-level noise when looking for a face in a photo. A failure to disregard irrelevant bits could lead to suboptimal generalization performance especially when the data contains spurious correlations. For instance, an image classifier that relies on background texture to identify objects is likely to fail when presented with a new image showing an object in an 'unusual' background (see, e.g., Refs [7, 30]). Understanding the principles behind the identification and extraction of relevant bits is therefore of fundamental and practical importance.

Formalizing this aspect of learning, the information bottleneck (IB) method provides a precise notion of relevance with respect to a prediction target: the relevant information in a source ($X$) is the bits that carry information about the target ($Y$) [26]. The relevant bits in $X$ are summarized in a representation ($Z$) via a stochastic map defined by an encoder $q(z|x)$, obeying the Markov constraint $Z \leftrightarrow X \leftrightarrow Y$.[1] In general a trade-off exists between the amount of discarded information (compression) and the remaining relevant information in $Z$ (prediction), thus motivating the IB cost function,[2]

$$L[q(z|x)] = I(Z; X) - \beta I(Z; Y), \tag{1}$$

where $\beta > 0$ denotes the trade-off parameter and $I(A; B)$ the mutual information. The first term favors succinct representations whereas the second encourages predictive ones. The IB loss is minimized by the representations that are most predictive of $Y$ at fixed compression, parametrized by the Lagrange multiplier $\beta$ (see, Fig 1a).

The IB method offers a highly versatile framework with wide-ranging applications, including neural coding [16], evolutionary population dynamics [22], statistical physics [9], clustering [25], deep

---

[1]This Markov chain implies $P_{Y|X,Z} = P_{Y|X}$ and $P_{Z|X,Y} = P_{Z|X}$ (see, e.g., Ref [6]).

[2]The optimization problem involving Eq (1) first appeared in a different context (see, e.g., Ref [27]).

35th Conference on Neural Information Processing Systems (NeurIPS 2021).

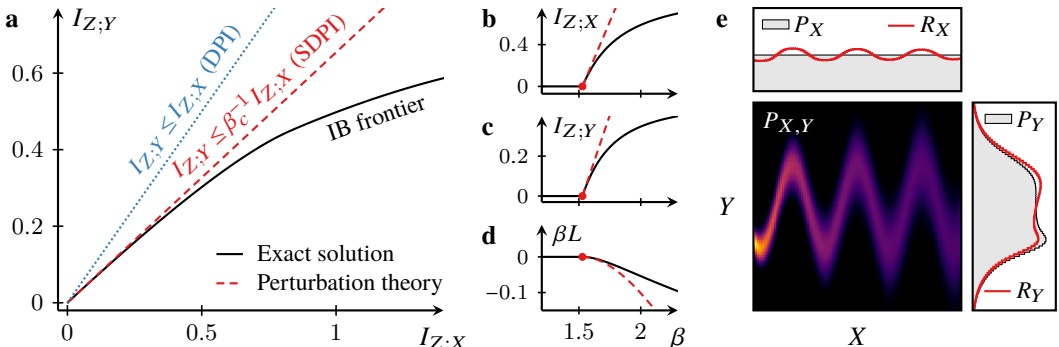

Figure 1: **Information bottleneck & Learning onset. a.** The IB frontier (solid) is parametrized by the trade-off parameter $\beta$ whose inverse is the slope of this curve. The relevant information is bounded from above by the data processing inequality (DPI) [dotted line] and its tight version, the strong data processing inequality (SDPI) [Eq (3), dashed line] which touches the IB curve at the origin. The slope at the origin is equal to the inverse critical trade-off parameter $\beta_c^{-1}$ which marks the learning onset (circles in (b-d)). **b-d.** Our controlled expansions (dashed) *vs* the exact solution (solid) for the joint distribution $P_{X,Y}$ shown in (e). The red curves in (e) depict the the perturbative IB encoder defined in Eq (14). We obtain the SDPI from Eqs (16) & (17) and the perturbative expansions in (b-d) from Eqs (26) & (27), see Appendix for relevant algorithms. Information is in bits.

learning [1–3] and reinforcement learning [10]. However the nonlinearity of the IB problem makes it computationally expensive and difficult to analyze, barring a few special cases [5]. This necessitates an investigation of tractable methods for solving the IB problem. The use of variational approximations to reduce the computational cost has paved the way for a massive scale-up of the IB method [3]. Complementing this approach, we report a new analytical result for the IB problem in the tractable limiting case of learning onset.

## 2   Learning Onset

Although the IB loss in Eq (1) favors a representation that encodes every relevant bit in $X$ when $\beta \to \infty$,[3] the optimal representation needs not contain any relevant information at finite $\beta$. To see this, we note that the loss vanishes for any uninformative representation $I(Z; X) = I(Z; Y) = 0$, and thus an informative representation yields a lower loss only when the relevant information in $Z$ is adequately large: a negative IB loss requires $I(Z; Y) > \beta^{-1} I(Z; X)$. But the relevant information is also bounded from above by the data processing inequality (DPI), $I(Z; Y) \leq I(Z; X)$, resulting from the the Markov constraint $Z \leftrightarrow X \leftrightarrow Y$ [6] (see, Fig 1a). Combining these inequalities yields

$$\beta^{-1} I(Z; X) < I(Z; Y) \leq I(Z; X), \tag{2}$$

which cannot be met when $\beta^{-1} > 1$. Hence the existence of an informative IB minimizer requires $\beta^{-1} \leq 1$. Indeed for any $P_{X,Y}$ with $I(X; Y) > 0$, there exists a critical trade-off parameter $\beta_c(X \to Y) \geq 1$ that marks the learning onset, separating two qualitatively distinct regimes: uninformative regime at $\beta < \beta_c$ and informative regime at $\beta > \beta_c$. The learning onset is the first in a series of transitions that emerges from the hierarchy of relevant information in the data [26].

Galvanized in part by the recent applications of the IB principle in deep learning, several works have attempted to characterized the IB transitions [8, 17, 28, 29]. However the IB problem remains intractable even in limiting cases and a complete characterization of the IB transitions remains elusive. In fact the only exception is the special case of Gaussian variables for which an exact solution exists [5]. In this work we derive a perturbation theory for the IB problem and offer the first complete description of the learning onset. We elaborate on the subtle differences between our theory and the previous works in Sec 6.

The learning onset is not only a special limit in the IB problem but also physically and practically relevant. It corresponds to the region where the relevant information per encoded bit is greatest and

---

[3]The compression term, while infinitesimally small in this limit, still penalizes irrelevant information and prefers a representation $Z$ that is the minimal sufficient statistics of $X$ for $Y$.

thus places a tight bound on the thermodynamic efficiency of predictive systems [23, 24]. An analysis the IB learning onset has recently found applications in statistical physics [9]. The (inverse) critical trade-off parameter is also a useful measure of correlation between two random variables [13]; indeed its square root satisfies all but the symmetry property of Rényi's axioms for statistical dependence measures [21]. Finally estimating the upper bound of $\beta_c$ might help weed out non-viable values of hyperparameters in deep learning techniques such as the variational information bottleneck [28, 29].

## 2.1 Strong data processing inequality

We can improve the bound on $\beta_c$ with the tight version of the DPI, the strong data processing inequality (SDPI) [4, 18, 20] (see, Fig 1a)

$$I(Z;Y) \leq \eta_{\mathrm{KL}}(X \to Y)I(Z;X) \tag{3}$$

where $\eta_{\mathrm{KL}}(X \to Y)$ denotes the contraction coefficient for the Kullback-Leibler divergence, defined via

$$\eta_{\mathrm{KL}}(X \to Y) \equiv \sup_{R_X \neq P_X} \frac{\mathrm{D_{KL}}(R_Y \| P_Y)}{\mathrm{D_{KL}}(R_X \| P_X)}. \tag{4}$$

Here $P_X$ and $P_Y$ denote the probability distributions of $X$ and $Y$. The supremum is over all allowed distributions given the space of $X$, and $R_Y$ is related to $R_X$ via the channel $P_{Y|X}$. Replacing the DPI with the SDPI in Eq (2), we obtain

$$\beta_c(X \to Y) \geq \eta_{\mathrm{KL}}(X \to Y)^{-1}. \tag{5}$$

In the following section we show that the equality holds, as expected (since the SDPI is tight). Note that $\eta_{\mathrm{KL}}(X \to Y)$ and $\beta_c(X \to Y)$ are generally asymmetric under $X \leftrightarrow Y$.

## 3 Perturbation Theory

We investigate the learning onset through the lens of perturbation theory. This method constructs the solution for a problem as a power series in a small parameter $\varepsilon$, when the solution for the limiting case $\varepsilon = 0$, the unperturbed solution, is accessible. For small $\varepsilon$, the higher order terms in this series represent ever smaller corrections to the unperturbed solution. To obtain these corrections, we insert the series solution into the initial problem and expand the resulting expressions as power series in $\varepsilon$, truncated at appropriate order. For example, the first-order theory drops all quadratic and higher terms (those proportional to $\varepsilon^2, \varepsilon^3, \dots$), resulting in a consistency condition for the linear correction (i.e., the term proportional to $\varepsilon$). Requiring consistency up to $\varepsilon^n$ leads to the $n^{\mathrm{th}}$-order perturbation theory. In practice the first few corrections suffice for a characterization of the problem in the vicinity of $\varepsilon = 0$.

Our theory is based on a controlled expansion around the critical trade-off parameter $\beta_c$ and some uninformative encoder $q_0(z|x) = q_0(z)$,

$$q(z|x) = q_0(z|x) + \varepsilon q_1(z|x) + \varepsilon^2 q_2(z|x) + \dots \tag{6}$$

$$I(Z;X) = \varepsilon I_{Z;X}^{(1)}[q_1] + \varepsilon^2 I_{Z;X}^{(2)}[q_1, q_2] + \dots, \tag{7}$$

where $\varepsilon \equiv \beta - \beta_c \to 0^+$ and $\sum_z q_n(z|x) = 0$ for $n \geq 1$ to ensure normalization. Note that $I_{Z;X}^{(0)}$ vanishes for uninformative $q_0$. The first and second-order informations capture the first and second-order growths of information as $\beta$ rises above $\beta_c$ and are given by (see Appendix for derivation)

$$I_{Z;X}^{(1)}[q_1] = \sum_x p(x) \sum_{z \in \mathcal{Z}_1} q_1(z|x) \ln \frac{q_1(z|x)}{q_1(z)} \tag{8}$$

$$I_{Z;X}^{(2)}[q_1, q_2] = \sum_x p(x) \Bigg( \sum_{z \in \mathcal{Z}_0} \frac{q_1(z|x)^2 - q_1(z)^2}{2q_0(z)} + \sum_{z \in \mathcal{Z}_1} q_2(z|x) \ln \frac{q_1(z|x)}{q_1(z)}$$
$$+ \sum_{z \in \mathcal{Z}_2} q_2(z|x) \ln \frac{q_2(z|x)}{q_2(z)} \Bigg), \tag{9}$$

where $\mathcal{Z}_0 = \mathrm{supp}(q_0)$ and $\mathcal{Z}_n = \mathrm{supp}(q_n) \setminus \bigcup_{i=0}^{n-1} \mathcal{Z}_i$ (i.e., $\mathcal{Z}_n$ contains representation classes or space that first appear in the support of the $n$th-order encoder).[4] The expansions for $q(z)$ and $q(z|y)$

---

[4] Our theory generalizes the expansions in Refs [28, 29] which considered the case $\mathcal{Z}_1 = \mathcal{Z}_2 = \emptyset$.

take the same form as Eq (6), and the expressions for $I(Z;Y)$ are identical to Eqs (7)-(9) but with $Y$ replacing $X$ everywhere. Finally we write down the loss function as a power series in $\varepsilon$,

$$L[q(z|x)] = \varepsilon L^{(1)}[q_1] + \varepsilon^2 L^{(2)}[q_1, q_2] + \dots, \tag{10}$$

where

$$L^{(1)}[q_1] = I^{(1)}_{Z;X}[q_1] - \beta_c I^{(1)}_{Z;Y}[q_1] \tag{11}$$

$$L^{(2)}[q_1, q_2] = I^{(2)}_{Z;X}[q_1, q_2] - \beta_c I^{(2)}_{Z;Y}[q_1, q_2] - I^{(1)}_{Z;Y}[q_1]. \tag{12}$$

## 3.1  First-order theory

Minimizing the first-order loss yields[5]

$$\min L^{(1)} = L^{(1)}[q_1^*] = 0 \quad \text{with} \quad \frac{q_1^*(z|x)}{q_1^*(z)} = \exp\left(\beta_c \sum_y p(y|x) \ln \frac{q_1^*(z|y)}{q_1^*(z)}\right) \text{ for } z \in \mathcal{Z}_1. \tag{13}$$

As the ratio $q_1(z|x)/q_1(z)$ does not depend on $z$, we eliminate the superfluous dependence on $z$ by defining

$$r(x) \equiv \frac{q_1^*(z|x)p(x)}{q_1^*(z)} \quad \text{for } z \in \mathcal{Z}_1, \quad \text{and} \quad r(y) \equiv \sum_x p(y|x)r(x). \tag{14}$$

Note that both $r(x)$ and $r(y)$ are non-negative and normalized: $\sum_x r(x) = \sum_y r(y) = 1$. Substituting Eqs (14) in (8) & (13), we obtain

$$I^{(1)}_{Z;X} = D_{KL}[r(x)\|p(x)] \sum_{z \in \mathcal{Z}_1} q_1^*(z)$$

$$I^{(1)}_{Z;Y} = D_{KL}[r(y)\|p(y)] \sum_{z \in \mathcal{Z}_1} q_1^*(z), \tag{15}$$

where

$$r(x) = p(x)e^{-\beta_c (D_{KL}[p(y|x)\|r(y)] - D_{KL}[p(y|x)\|p(y)])}. \tag{16}$$

Since the first-order loss vanishes [see, Eq (13)], we have $I^{(1)}_{Z;X}[q_1^*] - \beta_c I^{(1)}_{Z;Y}[q_1^*] = 0$ and thus

$$\beta_c = \frac{I^{(1)}_{Z;X}[q_1^*]}{I^{(1)}_{Z;Y}[q_1^*]} = \frac{D_{KL}[r(x)\|p(x)]}{D_{KL}[r(y)\|p(y)]}. \tag{17}$$

Note that an uninformative solution $r(x) = p(x)$ always satisfies Eq (16) and we must seek a nontrivial solution $r(x) \neq p(x)$.

We now show that the critical trade-off parameter is equivalent to the inverse contraction coefficient. First we note that $r(x)$ in Eq (16) is a solution to a different optimization, described by a loss function $\mathcal{L}[f] = D_{KL}[f(x)\|p(x)] - \beta_c D_{KL}[f(y)\|p(y)]$. That is, $\delta\mathcal{L}/\delta f|_{f \to r} = 0$ and $\min \mathcal{L} = \mathcal{L}[r] = 0$. It follows immediately that $\delta\left(\frac{D_{KL}[f(y)\|p(y)]}{D_{KL}[f(x)\|p(x)]}\right)/\delta f|_{f \to r} = 0$ for $D_{KL}[r(x)\|p(x)] > 0$, therefore

$$\beta_c^{-1} = \frac{D_{KL}[r(y)\|p(y)]}{D_{KL}[r(x)\|p(x)]} = \sup_{f \neq p} \frac{D_{KL}[f(y)\|p(y)]}{D_{KL}[f(x)\|p(x)]} = \eta_{KL}(X \to Y), \tag{18}$$

where the first and last equalities come from Eqs (17) & (4), respectively. The above analysis provides an alternative derivation of the equivalence between the contraction coefficients of mutual information and KL divergence [4, 18].

While our first-order theory provides a method for identifying the critical trade-off parameter by solving Eqs (16) & (17), it is incomplete. The optimal encoder in Eq (13) is determined only up to a multiplicative factor. Consequently the informations in Eq (15) still depend on $q_1(z)$ which can take any positive value (for $z \in \mathcal{Z}_1$). This unphysical scale invariance is broken in the second-order theory.

---

[5]Unlike in the original IB problem, here the optimization is unconstrained since the normalization $\sum_z q_1(z|x) = 0$ sums over both $\mathcal{Z}_0$ and $\mathcal{Z}_1$, and only the latter enters our first-order theory.

## 3.2 Second-order theory

From Eqs (9) & (12), we write down the second-order loss

$$L^{(2)}[q_1, q_2] = \sum_{z \in \mathcal{Z}_0} \frac{\sum_{x,x'} q_1(z|x) K(x, x') q_1(z|x')}{2q_0(z)} - I_{Z;Y}^{(1)}[q_1] \tag{19a}$$

$$+ \sum_x p(x) \sum_{z \in \mathcal{Z}_1} q_2(z|x) \left( \ln \frac{q_1(z|x)}{q_1(z)} - \beta_c \sum_y p(y|x) \ln \frac{q_1(z|y)}{q_1(z)} \right) \tag{19b}$$

$$+ \sum_x p(x) \sum_{z \in \mathcal{Z}_2} q_2(z|x) \left( \ln \frac{q_2(z|x)}{q_2(z)} - \beta_c \sum_y p(y|x) \ln \frac{q_2(z|y)}{q_2(z)} \right), \tag{19c}$$

where we define

$$K(x, x') \equiv \delta(x, x') p(x) + (\beta_c - 1) p(x) p(x') - \beta_c \sum_y p(y) p(x|y) p(x'|y). \tag{20}$$

Optimizing $L^{(2)}$ with respect to $q_2$ (for $\mathcal{Z}_1$ and $\mathcal{Z}_2$ separately) results in stationary conditions, which equate the terms in the parentheses of Eqs (19b) & (19c) to zero.[6] Eliminating $I_{Z;Y}^{(1)}$ in Eq (19a) with Eq (15), we have

$$L^{(2)}[q_1] = -\mathrm{D_{KL}}[r(y)\|p(y)] \sum_{z \in \mathcal{Z}_1} q_1^*(z) + \sum_{z \in \mathcal{Z}_0} \frac{\sum_{x,x'} q_1(z|x) K(x, x') q_1(z|x')}{2q_0(z)}. \tag{21}$$

Minimizing this loss function with respect to $q_1$ and subject to the normalization $\sum_z q_1(z|x) = 0$ gives

$$\sum_{x'} K(x, x') \frac{q_1^*(z|x')}{q_0(z)} = - \left( \sum_{z' \in \mathcal{Z}_1} q_1^*(z') \right) \sum_{x'} K(x, x') \frac{r(x')}{p(x')} \quad \text{for } z \in \mathcal{Z}_0. \tag{22}$$

Substituting the above in Eq (21) leads to

$$L^{(2)}[q_1^*] = -\mathrm{D_{KL}}[r(y)\|p(y)] \sum_{z \in \mathcal{Z}_1} q_1^*(z) + \frac{\kappa}{2} \left( \sum_{z \in \mathcal{Z}_1} q_1^*(z) \right)^2, \tag{23}$$

where we define

$$\kappa \equiv \sum_{x,x'} \frac{r(x) K(x, x') r(x')}{p(x) p(x')}. \tag{24}$$

Assuming $\kappa > 0$,[7] the final minimization with respect to $\sum_{z \in \mathcal{Z}_1} q_1(z)$ yields

$$\sum_{z \in \mathcal{Z}_1} q_1^*(z) = \frac{1}{\kappa} \mathrm{D_{KL}}[r(y)\|p(y)], \tag{25}$$

$$L^{(2)}[q^*] = -\frac{1}{2\kappa} \mathrm{D_{KL}}[r(y)\|p(y)]^2. \tag{26}$$

Finally we eliminate the remaining dependence on $q_1$ in Eq (15) and write down the first-order information

$$I_{Z;X}^{(1)} = \frac{1}{\kappa} \mathrm{D_{KL}}[r(x)\|p(x)] \mathrm{D_{KL}}[r(y)\|p(y)]$$

$$I_{Z;Y}^{(1)} = \frac{1}{\kappa} \mathrm{D_{KL}}[r(y)\|p(y)]^2. \tag{27}$$

We see that the second-order perturbation theory fixes the scales of the leading corrections to mutual information, thus completing our analysis of the learning onset. Furthermore these leading corrections are related via $I_{Z;X}^{(1)} = \beta_c I_{Z;Y}^{(1)}$ and $L^{(2)} = -I_{Z;Y}^{(1)}/2$.

---

[6]This optimization is unconstrained since the second-order loss does not depend on $q_2$ with $z \in \mathcal{Z}_0$ (see, footnote 5). The resulting stationary conditions are identical to Eq (13) for $q_1$ with $z \in \mathcal{Z}_1$ and $q_2$ with $z \in \mathcal{Z}_2$.

[7]For $\kappa \leq 0$, the loss function in Eq (23) is unbounded from below and a higher order perturbation theory is required to fix the scale of $q_1$.

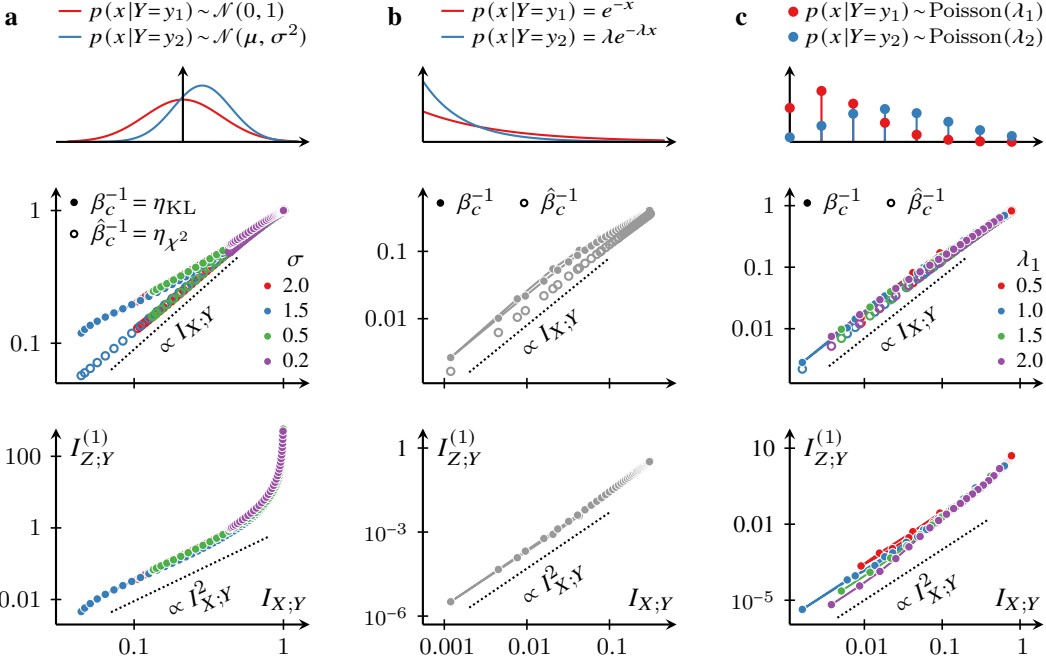

Figure 2: **Learning onset in binary classification.** We illustrate the results of our theory for the case of a binary target variable with equal probability assigned to each class, i.e., $Y \in \{y_1, y_2\}$ and $p(Y=y_1) = p(Y=y_2) = 1/2$, for three different sets of conditional distributions $p(x|y)$ (a-c, top row). **a.** The source data $X$ are drawn from a Gaussian distribution whose mean and variance depend on $Y$ (top panel). We set the mean to zero and variance to one for $Y = y_1$ and solve the IB learning onset for various values of mean $\mu$ and variance $\sigma$ for $Y = y_2$. The middle panel depict the critical trade-off parameter, predicted by our theory in Sec 3 (filled circles) and the methods from previous works described in Sec 6 (empty circles). The bottom panel shows the information response to a small perturbation in trade-off parameter [for definition see, Eq (7)]. The theory predictions are plotted against the data mutual information, parametrized by the mean $\mu$ of $p(x|y_2)$ for four different values of standard deviations (see legend). The dotted lines display the power dependence and serves only as a guide to the eye to aid comparisons. **b.** Same as (a) but for exponential distributions and the curves are parametrized by the rate parameter $\lambda$ of the exponential distributions (see, top panel). **c.** Same as (a) but for Poisson distributions and the curves are parametrized by $\lambda_2$ [mean of $p(x|y_2)$] for four values of $\lambda_1$ [mean of $p(x|y_1)$] (see legend). Information is in bits.

## 4  Numerical Results

We now turn to comparing our theory to numerical results. In Fig 1, we compare the results from our perturbation theory [Eqs (16), (17), (26) & (27)] to the numerically exact solution of the IB problem for a synthetic joint distribution (shown in Fig 1e). Our theory correctly identifies the critical trade-off parameter and captures the leading corrections to the mutual information and IB loss in the vicinity of the learning onset (see, Fig 1b-d). The inverse critical trade-off parameter $\beta_c^{-1}$ coincides with the slope of the strong data processing inequality (SDPI) which provides a tight upper bound for the IB frontier (Fig 1a). Note that the SDPI is tight at the origin [$I(Z; X) = I(Z; Y) = 0$] and is therefore fully characterized by our analysis of the learning onset.

**Binary classification**  In Fig 2, we consider the onset of learning for binary classification in which the target $Y$ is a binary random variable with equal probability for each class and the source variable $X$ is drawn from a distribution that depends on the realization of $Y$. In other words, provided with some data $x$, we ask whether it was drawn from blue or red distributions in the top panel of Fig 2. In all cases we see a general trend that the inverse critical trade-off parameter $\beta_c^{-1}$ and the relevant information response $I_{Z;Y}^{(1)}$ increase with available information $I(X; Y)$. Indeed for the Gaussian case (Fig 2a), the information response diverges in the high information limit (equivalent to a large difference in

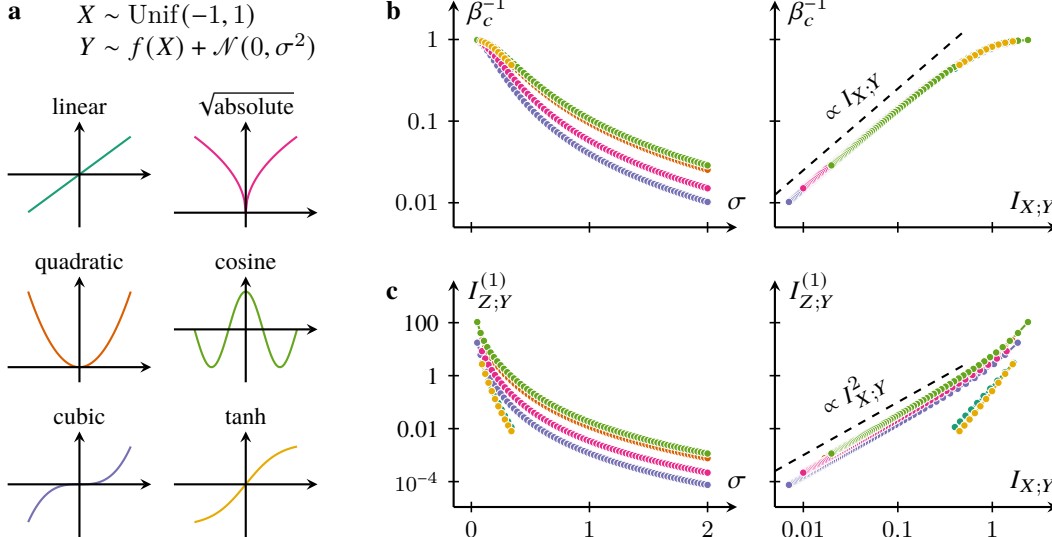

Figure 3: **Learning onset for noisy functional relationships. a.** Functions used in data generation: $X \sim \text{Unif}(-1, 1)$ and $Y \sim f(X) + \mathcal{N}(0, \sigma^2)$. **b.** The inverse critical trade-off parameter $\beta_c^{-1}$ *vs* noise level parametrized by the noise standard deviation $\sigma$ (left) and by available information $I(X;Y)$ (right). **c.** The first-order growth of information *vs* noise level parametrized by $\sigma$ (left) and by $I(X;Y)$ (right). Both the maximum relevant information per extract bit $\beta_c^{-1}$ and the first-order relevant information $I_{Z;Y}^{(1)}$ decrease with noise level as it becomes increasingly difficult to extract relevant information. The dashed lines display the power dependence and serves only as a guide to the eye to aid comparisons. Information is in bits.

the means of the Gaussian distributions) which is also the limit where binary classification becomes deterministic, $I(X;Y) \to 1$ bit.

**Noise dependence** In Fig 3, we depict the critical trade-off parameter and information response for joint distributions generated from $X \sim \text{Unif}(-1, 1)$ and $Y \sim f(X) + \mathcal{N}(0, \sigma^2)$ for various functional associations (Panel a). We see that the critical trade-off tends to one in the low noise limit, as expected for a deterministic functional relationship [14]. At higher noise level, $\beta_c$ increases with $\sigma$ as it becomes harder to extract relevant bits from the data. This fact is also reflected in the first-order information $I_{Z;Y}^{(1)}$ which measures the change in relevant information as the trade-off parameter $\beta$ exceeds the critical value. For all functions considered, $I_{Z;Y}^{(1)}$ decreases with increasing noise standard deviation. Interestingly we see that the information response diverges in the deterministic limit similar to the binary classification example shown in Fig 2a. Note that $I_{Z;X}^{(1)} = \beta_c I_{Z;Y}^{(1)}$ and $L^{(2)} = -I_{Z;Y}^{(1)}/2$ [see Eqs (26) & (27)].

## 5 Learning Onset for Gaussian Variables

At first sight it seems that our theory, which is agnostic about the discrete or continuous nature of the representation, is at odd with the exact solution for Gaussian variables which is based on a continuous representation [5]. In this section we show that our theory captures the learning onset for joint Gaussian variables. Importantly we demonstrate that a discrete representation of continuous variables can describe the learning onset just as well as continuous ones.

Consider joint Gaussian variables

$$\begin{bmatrix} X \\ Y \end{bmatrix} \sim \mathcal{N}\left( \begin{bmatrix} 0 \\ 0 \end{bmatrix}, \begin{bmatrix} \Sigma_X & \Sigma_{XY} \\ \Sigma_{YX} & \Sigma_Y \end{bmatrix} \right). \tag{28}$$

A convenient ansatz for $r(x)$ and $r(y)$ [for definitions, see, Eq (14)] is a Gaussian distribution,

$$R_X = \mathcal{N}(\nu_X, \Lambda_X) \quad \text{and} \quad R_Y = \mathcal{N}(\nu_Y, \Lambda_Y). \tag{29}$$

where $(\nu_X, \Lambda_X)$ denotes the mean vector and covariance matrix for $R_X$ and $(\nu_Y, \Lambda_Y)$ for $R_Y$. Using this ansatz, we write down the KL divergences in the exponent of Eq (16),

$$D_{\mathrm{KL}}[p(y|x)\|r(y)] = \frac{1}{2}\left((\mu_{Y|x} - \nu_Y)^\mathsf{T}\Lambda_Y^{-1}(\mu_{Y|x} - \nu_Y) + \mathrm{tr}[\Lambda_Y^{-1}\Sigma_{Y|X}] - d_Y + \ln\frac{|\Lambda_Y|}{|\Sigma_{Y|X}|}\right) \quad (30)$$

$$D_{\mathrm{KL}}[p(y|x)\|p(y)] = \frac{1}{2}\left(\mu_{Y|x}^\mathsf{T}\Sigma_Y^{-1}\mu_{Y|x} + \mathrm{tr}[\Sigma_Y^{-1}\Sigma_{Y|X}] - d_Y + \ln\frac{|\Sigma_Y|}{|\Sigma_{Y|X}|}\right), \quad (31)$$

where $\Sigma_{Y|X} = \Sigma_Y - \Sigma_{YX}\Sigma_X^{-1}\Sigma_{XY}$, $d_Y$ denotes the dimensionality of $Y$ and we define $\mu_{Y|x} \equiv \Sigma_{YX}\Sigma_X^{-1}x$. The ratio between $r(x)$ and $p(x)$ is given by

$$\ln\frac{r(x)}{p(x)} = \frac{1}{2}\left(-(x - \nu_X)^\mathsf{T}\Lambda_X^{-1}(x - \nu_X) + x^\mathsf{T}\Sigma_X^{-1}x + \ln\frac{|\Sigma_X|}{|\Lambda_X|}\right). \quad (32)$$

Since Eqs (30)-(32) are related via Eq (16) which holds for all values of $x$, we take the logarithm of Eq (16) and equate the terms quadratic in $x$, linear in $x$ and constants separately, yielding

$$\Lambda_X^{-1} - \Sigma_X^{-1} = \beta_c\Sigma_X^{-1}\Sigma_{XY}(\Lambda_Y^{-1} - \Sigma_Y^{-1})\Sigma_{YX}\Sigma_X^{-1} \quad (33)$$

$$\Lambda_X^{-1}\nu_X = \beta_c\Sigma_X^{-1}\Sigma_{XY}\Lambda_Y^{-1}\nu_Y \quad (34)$$

$$\nu_X^\mathsf{T}\Lambda_X^{-1}\nu_X = \ln\frac{|\Sigma_X|}{|\Lambda_X|} + \beta_c\left(\nu_Y^\mathsf{T}\Lambda_Y^{-1}\nu_Y + \mathrm{tr}[(\Lambda_Y^{-1} - \Sigma_Y^{-1})\Sigma_{Y|X}] - \ln\frac{|\Sigma_Y|}{|\Lambda_Y|}\right). \quad (35)$$

We can find a solution to this set of equations by letting $\Lambda_X = \Sigma_X$ (which also leads to $\Lambda_Y = \Sigma_Y$). For this choice of covariance matrix, both sides of Eq (33) vanish and Eqs (34) & (35) reduce to[8]

$$\left(1 - \beta_c(1 - \Sigma_{X|Y}\Sigma_X^{-1})\right)\nu_X = 0. \quad (36)$$

Solving the above eigenproblem for the smallest possible critical trade-off parameter, we find $\beta_c = (1 - \lambda_{\min})^{-1}$ and $\nu_X \propto \phi_{\min}$ where $\lambda_{\min}$ denotes the smallest eigenvalue of $\Sigma_{X|Y}\Sigma_X^{-1}$ and $\phi_{\min}$ the corresponding eigenvector. While both [5] and our work identify the same critical trade-off parameter and reveal the importance of the spectrum of $\Sigma_{X|Y}\Sigma_X^{-1}$, the analyses are distinct in that the representation is continuous in [5] but can be discrete in our theory.[9]

## 6    Comparisons to Previous Works

The recent applications of the IB principle in machine learning [1–3, 7] have sparked much interest in characterizing the structure of the IB problem [28, 29]. Several works underscore the learning onset and IB transitions as important limiting cases, not least because they are a direct manifestation of the hierarchical structure of the relevant information in the data [5, 8, 17, 28, 29]. However the attempts to derive a perturbation theory for the learning onset are plagued by a flawed assumption that the representation space does not expand beyond the support of the unperturbed, uninformative encoder [8, 28, 29]. Equivalent to setting $\mathcal{Z}_1 = \mathcal{Z}_2 = \emptyset$ in Eqs (8) & (9) in our theory, this assumption significantly simplifies the analysis but the resulting theory generally fails to identify the critical trade-off parameter.[10] This raises serious questions about the insights gleaned from such expansions around a seemingly arbitrary point. In the following we explore the differences between our full treatment and the perturbation theory derived in previous works. In particular we argue that the theory in previous work describes the learning onset of a non-standard IB problem, defined with $\chi^2$–information (instead of Shannon information).

---

[8]Equation (35) becomes the same as Eq (34) but with $\nu_X^\mathsf{T}\Sigma_X^{-1}$ multiplied from the left.

[9]We can always choose the unperturbed encoder to be an all-to-one map ($q_0(z_0|x) = 1$) and let the linear correction have access to one additional alphabet ($q_1(z_1|x) > 0$).

[10]Our set-up differs slightly from Refs [28, 29] in that we ask how optimal encoders respond to a small change in $\beta$ as opposed to how the loss function changes in response to a small perturbation to an encoder. However this difference is not the reason why our theory produces a tight bound on the learning onset. Allowing the representation to take values outside the support of the unperturbed encoder is key to capturing the learning onset regardless of how a perturbation theory is constructed.

Setting $\mathcal{Z}_1 = \mathcal{Z}_2 = \emptyset$, the leading correction to the IB loss is of second order and is given by the first term of Eq (19a),[11]

$$L^{(2)}[q_1] = \sum_{z \in \mathcal{Z}_0} \frac{\sum_{x,x'} q_1(z|x) K(x,x') q_1(z|x')}{2 q_0(z)}, \tag{37}$$

where the dependence on $\beta_c$ is implicit [see, Eq (20), for the definition of $K(x,x')$]. We see that $K(x,x')$ is the Hessian of the loss function and its eigenvalues determine the curvatures of the loss landscape in the vicinity of the unperturbed encoder. In this theory the learning onset corresponds to the emergence of a direction along which the loss decreases quadratically—i.e., when the smallest eigenvalue first becomes negative. Note that $K(x,x')$ always has a vanishing eigenvalue, resulting from the fact that all uninformative perturbations $q_1(z|x) = q_1(z)$ lead to the same loss.[12] In practice we may identify the learning onset with the point where the second smallest eigenvalue becomes zero but a more efficient method exists, see below. Similarly to our first-order theory (Sec 3.1), this eigenvalue problem yields only the direction of the first-order encoder and a higher order theory is required to fix the scale.

It is worth pointing out that if we define the IB problem [Eq (1)] with $\chi^2$–information instead of the standard Shannon information,[13] Eq (37) is identical (up to a multiplicative factor) to the first-order loss in our full treatment (i.e., with $\mathcal{Z}_1 \neq \emptyset$). Indeed the resulting learning onset coincides with the SDPI for $\chi^2$–information. The contraction coefficient for $\chi^2$–information, $\eta_{\chi^2}$, is exactly the squared maximal correlation (for a review, see, e.g., Ref [15]) and is therefore symmetric under $X \leftrightarrow Y$ and equal to the square of the second largest singular value of the divergence transition matrix [12, 21],

$$B(x,y) \equiv \frac{p(x,y)}{\sqrt{p(x)p(y)}} \quad \text{for} \quad p(x)p(y) > 0, \quad \text{and} \quad B(x,y) \equiv 0 \quad \text{otherwise.} \tag{38}$$

Finally we note that $\eta_{\chi^2} \leq \eta_{\mathrm{KL}}$ [19, 20], hence the perturbation theory based on fixed representation space gives an upper bound to the critical trade-off parameter of the standard IB problem.

Figure 2 demonstrates that even for simple binary classification, the theory with fixed representation space, which predicts $\hat{\beta}_c = \eta_{\chi^2}^{-1}$ (empty circles), does not correctly identify the learning onset (filled circles). For the set of examples shown, we see that the discrepancy between $\beta_c$ and $\hat{\beta}_c$ is greatest for the Gaussian case and at lower available information. Note that in the deterministic limit [$I(X;Y) = 1$ bit for binary classification] all contraction coefficients tend to one and we do not expect any discrepancy there.

# 7   Discussion & Outlook

We derive a perturbation theory for the IB problem and offer a glimpse of the intimate connections between the learning onset and the strong data processing inequality. In future works we aim to build on our results to develop an algorithm for estimating the contraction coefficient from samples and explore novel methods for solving the IB problem in this limit. It would be interesting to further leverage the wealth of rigorous results from the literature on hypercontractivity and strong data processing inequalities to better understand the learning onset in the IB problem. In addition, various numerical techniques developed for the IB problem could significantly extend the range of applicability of contraction coefficients.

In Sec 5, we show that a discrete representation can also capture the learning onset for Gaussian variables. Our approach contrasts with the exact solution of Ref [5] which uses continuous representation. This highlights the degeneracy of the global minimum in the IB problem and implies that discrete representations of continuous variables needs not be suboptimal.

While the IB problem formulated with Shannon information is somewhat unique [11], our work reveals that the analyses of the learning onset would be much simplified if one were to define the IB loss with $\chi^2$–information instead of Shannon information. The IB principle based on other $f$–information could provide a more tractable formulation for certain problems and offer an insight not readily available otherwise.

---

[11]Note that this loss depends only on the first-order encoder $q_1$. The second-order encoder $q_2$ appears in higher order theories.

[12]It is easy to verify that $\sum_{x'} K(x,x') = 0$.

[13]The $\chi^2$–information is defined as follows, $I_{\chi^2}(X;Y) \equiv \sum_{x,y} p(x)p(y) \left( \frac{p(x,y)}{p(x)p(y)} - 1 \right)^2$

## Acknowledgments and Disclosure of Funding

We thank Shervin Parsi and Sarang Gopalakrishnan for useful discussions during the early stages of the project. This work was supported in part by the National Institutes of Health BRAIN initiative (R01EB026943), the National Science Foundation, through the Center for the Physics of Biological Function (PHY-1734030), the Simons Foundation and the Sloan Foundation.

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
