# Perturbation Theory for the Information Bottleneck Appendix

**Vudtiwat Ngampruetikorn, David J. Schwab**
Initiative for the Theoretical Sciences, The Graduate Center, CUNY

## A  Power series expansion of information

To derive Eqs (8) & (9), we first write down the encoder as a power series

$$q(z|x) = q_0(z) + \varepsilon q_1(z|x) + \varepsilon^2 q_2(z|x) + O(\varepsilon^3)$$

where we use the fact that $q(z|x) = q(z)$ at $\varepsilon = 0$ and $O(\varepsilon^3)$ denotes terms of order three and above. Marginalizing out $x$ gives

$$q(z) = \sum_x p(x)q(z|x) = q_0(z) + \varepsilon q_1(z) + \varepsilon^2 q_2(z) + O(\varepsilon^3), \text{ where } q_n(z) = \sum_x p(x)q_n(z|x).$$

Using these equations, we expand the following expression as a power series in $\varepsilon$ (up to $\varepsilon^2$),

$$q(z|x) \ln \frac{q(z|x)}{q(z)} = \left( q_0(z) + \varepsilon q_1(z|x) + \varepsilon^2 q_2(z|x) + O(\varepsilon^3) \right) \ln \frac{q_0(z) + \varepsilon q_1(z|x) + \varepsilon^2 q_2(z|x) + O(\varepsilon^3)}{q_0(z) + \varepsilon q_1(z) + \varepsilon^2 q_2(z) + O(\varepsilon^3)}$$

$$= \begin{cases} \begin{aligned} & \varepsilon(q_1(z|x) - q_1(z)) \\ & \quad + \varepsilon^2 \left( \frac{(q_1(z|x) - q_1(z))^2}{2q_0(z)} + q_2(z|x) - q_2(z) \right) + O(\varepsilon^3) \end{aligned} & \text{if } q_0(z) > 0 \\[4pt] \begin{aligned} & \varepsilon q_1(z|x) \ln \frac{q_1(z|x)}{q_1(z)} \\ & \quad + \varepsilon^2 \left( q_2(z|x) \ln \frac{q_1(z|x)}{q_1(z)} + q_2(z|x) - \frac{q_1(z|x)q_2(z)}{q_1(z)} \right) + O(\varepsilon^3) \end{aligned} & \text{if } q_0(z) = 0 \text{ and } q_1(z) > 0 \\[4pt] \varepsilon^2 q_2(z|x) \ln \frac{q_2(z|x)}{q_2(z)} + O(\varepsilon^3) & \text{if } q_0(z) = q_1(z) = 0 \text{ and } q_2(z) > 0 \\[4pt] O(\varepsilon^3) & \text{if } q_0(z) = q_1(z) = q_2(z) = 0 \end{cases}$$

We now define

$$\mathcal{Z}_n \equiv \{ z \mid q_n(z) > 0 \text{ and } q_m(z) = 0 \text{ for } 0 \le m < n \} \quad \text{and} \quad \mathcal{Z}_0 \equiv \mathrm{supp}(q_0).$$

Finally the power series for the mutual information is given by (up to $\varepsilon^2$),

$$I(Z;X) = \sum_x p(x) \sum_z q(z|x) \ln \frac{q(z|x)}{q(z)}$$

$$= \sum_x p(x) \sum_{z \in \mathcal{Z}_0} \left( \varepsilon(q_1(z|x) - q_1(z)) + \varepsilon^2 \left( \frac{(q_1(z|x) - q_1(z))^2}{2q_0(z)} + q_2(z|x) - q_2(z) \right) + O(\varepsilon^3) \right)$$

$$+ \sum_x p(x) \sum_{z \in \mathcal{Z}_1} \left( \varepsilon q_1(z|x) \ln \frac{q_1(z|x)}{q_1(z)} + \varepsilon^2 \left( q_2(z|x) \ln \frac{q_1(z|x)}{q_1(z)} + q_2(z|x) - \frac{q_1(z|x)q_2(z)}{q_1(z)} \right) + O(\varepsilon^3) \right)$$

$$+ \sum_x p(x) \sum_{z \in \mathcal{Z}_2} \left( \varepsilon^2 q_2(z|x) \ln \frac{q_2(z|x)}{q_2(z)} + O(\varepsilon^3) \right) + O(\varepsilon^3)$$

$$= \varepsilon \sum_x p(x) \sum_{z \in \mathcal{Z}_1} q_1(z|x) \ln \frac{q_1(z|x)}{q_1(z)}$$

$$+ \varepsilon^2 \sum_x p(x) \left( \sum_{z \in \mathcal{Z}_0} \frac{(q_1(z|x) - q_1(z))^2}{2q_0(z)} + \sum_{z \in \mathcal{Z}_1} q_2(z|x) \ln \frac{q_1(z|x)}{q_1(z)} + \sum_{z \in \mathcal{Z}_2} q_2(z|x) \ln \frac{q_2(z|x)}{q_2(z)} \right)$$

$$+ O(\varepsilon^3)$$

35th Conference on Neural Information Processing Systems (NeurIPS 2021).

where we used the fact that $\sum_x p(x) q_n(z|x) = q_n(z)$. Equating the terms in this equation to that of Eq (7) leads directly to Eqs (8) & (9)

## B  Algorithms for the information bottleneck and learning onset

---

**Algorithm 1:** Information bottleneck method [26]

---

**Input** : $p(x,y)$, $\beta \geq 1$ and tolerance $\delta$
**Output :** IB encoder $q(z|x)$

Initialize $q(z|x)$ such that $q(z|x) > 0$ and $\sum_z q(z|x) = 1$

**repeat**

$\quad | \quad \tilde{q}(z|x) \leftarrow q(z|x)$
$\quad | \quad q(z) \quad \leftarrow \sum_x q(z|x) p(x)$
$\quad | \quad q(y|z) \leftarrow \sum_x q(z|x) p(x,y)/q(z)$
$\quad | \quad q(z|x) \leftarrow q(z) \exp\{-\beta \, \mathrm{D_{KL}}[p(y|x)\|q(y|z)]\}$
$\quad | \quad q(z|x) \leftarrow q(z|x)/\sum_{z'} q(z'|x)$

**until** $\|q(z|x) - \tilde{q}(z|x)\| < \delta$

---

**Algorithm 2:** IB learning onset

---

**Input** : $p(x,y)$ and tolerances $(\delta, \epsilon)$
**Output :** Critical trade-off parameter $\beta_c$ and perturbative encoders $r(x)$ $\qquad$ ▷ see Eq (14)

**repeat**

$\quad | \quad$ Initialize $[r(x), \beta_c]$ (randomly) such that $r(x) > 0$ and $\beta_c > 1$
$\quad | \quad$ **repeat**
$\quad | \quad | \quad [r_0(x), \beta_0] \leftarrow [r(x), \beta_c]$
$\quad | \quad | \quad r(x) \qquad \leftarrow r(x)/\sum_{x'} r(x')$
$\quad | \quad | \quad r(y) \qquad \leftarrow \sum_x p(y|x) r(x)$ $\qquad$ ▷ Eq (14)
$\quad | \quad | \quad \beta_c \qquad \leftarrow \mathrm{D_{KL}}[r(x)\|p(x)]/\mathrm{D_{KL}}[r(y)\|p(y)]$ $\qquad$ ▷ Eq (17)
$\quad | \quad | \quad r(x) \qquad \leftarrow q(z) \exp\{-\beta_c (\mathrm{D_{KL}}[p(y|x)\|r(y)] - \mathrm{D_{KL}}[p(y|x)\|p(y)])\}$ $\quad$ ▷ Eq (16)
$\quad | \quad$ **until** $\|[r(x), \beta_c] - [r_0(x), \beta_0]\| < \delta$

**until** $\|r(x) - p(x)\| > \epsilon$ $\qquad$ ▷ To avoid uninformative solution $r(x) = p(x)$

---