# OpenReview forum: "Perturbation Theory for the Information Bottleneck"
_NeurIPS.cc/2021/Conference — NeurIPS 2021 Poster_

### Official Review · Reviewer_o32a · 2021-07-15

**Rating:** 7
**Confidence:** 3

**Summary:**

The authors investigate the learning onset for the information bottleneck method, i.e., the behavior of the problem at \beta being close to the critical parameter \beta_c. More specifically, the IB curve is studied via a perturbation around the optimal compression map. In contrast to previous work, the authors allow this perturbation also outside of the alphabet of the optimal compression.

**Limitations And Societal Impact:**

The authors mention that the results concern only the learning onset, which is sufficient in my opinion. I also agree with the authors that no societal impacts are to be anticipated from this theoretical work.

**Main Review:**

The manuscript is well-written and seems mathematically correct (although I have to admit that I did not check every single detail). Understanding the properties of such an important and popular optimization problem is relevant in its own right. The insights are thus interesting and relevant, albeit only for a very small community (e.g., while the IB framework is often used for training neural networks, it is rarely the IB functional that is directly optimized, but some variational (or other) approximations). I am therefore generally in favor of this paper.

There are a few consideration that I would like to add:
- In equations (8)-(9) it is not clear whether these are definitions or calculations. If they are calculations, I would appreciate to see these carried out in an appendix. These two equations are central to the paper, as the remaining derivations start from these assumptions.
- In line 100 you claim that the ratio q1(z|x)/q1(z) does not depend on z. This is not obvious to me. It may very well be that I misunderstood some central part of the paper, but then it may be that this also escapes the attention of the occasional reader. May I ask you to expand on this statement a bit?
- In equations (21)-(23) it is not exactly clear where and why the optimum q1* should be used. (And I assume that q1* is the optimizer of Section 3.1.) For example, why is q1* used in (21), but not in (22) and (23)? I would expect that q1* should at least be used in (23), as in its left-hand side of the equation.
- In (22) is the r(x') defined via (14) or via (16), or is this the same by selecting q1(z|x)=q1*(z|x) as in (13)? This also leaves unclear to me whether in (15) the sums should run over q1* instead of q1, as (15) is obtained via inserting (13) into (14), and (13) uses the minimizer q1*. Again, I may have missed an important point here, in which case I apologize.
- In Fig. 2 you add the proportionality to I(X;Y) in the figures. Are these proportionalities derived from or consistent with theory? I did not see this clearly mentioned in the paper. Even then, it is peculiar that the Gaussian case exhibits a bigger difference between the proportionality and the numerical results than previous approaches yielded. I would appreciate if you could expand on this fact somewhere in the text.
- In lines 161-163 you connect the leading corrections for I(Z;Y) and I(Z;X), and you justify this via equations (26)-(27). If these equations hold true in general, then I would appreciate to see lines 161-163 in the general part of the paper (e.g., Section 3), rather than just at the end of the numerical experiments.
- In line 165 you mention that the theory is agnostic about the discrete or continuous nature of the representation, while in line 184 you claim that the representations are discrete in this work. Which of the two statements is correct? They appear to be in conflict, at least to me. More generally, you claim in lines 183-184 that the analyses are distinct. I wonder if, using a Gaussian assumption, the functions r(x) and r(y) would have the same shape as they do in the discrete case. More generally, where is the discreteness of your assumption essential, and where can it be generalized to continuous distributions? I acknowledge that this last question is a very complicated one and I would fully accept as answer that this investigation is out of scope. However, I would be willing to share an educated guess here in the replies, I would read it with interest.


Finally, there are some minor comments that I have, but that did not influence the score of the review:
- in lines 82-83, I think you mean that the corrections get larger if \epsilon deviates from zero, right?
- in line 91, the notation \delta_n,0 is not fully clear. I prefer the notation in eq. (20) or in line 126.

**Time Spent Reviewing:**

3

---

> ### Author Response · Authors · 2021-08-10
> **Author Response**
>
> Thank you for your thorough reading and helpful comments and suggestions. We respond to your questions in turn below.
>
> ---
> > _In equations (8)-(9) it is not clear whether these are definitions or calculation._
>
> We derive Eqs (8) and (9) by inserting the series solutions [Eqs (6-7)] into the definition of mutual information and enforcing consistency up to the first and second orders in $\epsilon$, respectively. We will clarify this, and the derivation will be included in the Appendix of the updated manuscript.
>
> &nbsp;
> > _In line 100 you claim that the ratio q1(z|x)/q1(z) does not depend on z._
>
> To see this result, we substitute $q_1(z|y)=\sum_{x'}q_1(z|x')p(x'|y)$ into the *rhs* of Eq (13),
> $$
> \frac{q_1^*(z|x)}{q_1^*(z)}
> =
> \exp\left(
>     \beta_c\sum\nolimits_yp(y|x)\ln\left(\sum\nolimits_{x'}p(x'|y)
>     \frac{q_1^*(z|x')}{q_1^*(z)}\right)
> \right)
> $$
> Since $q_1^*(z|x)/q_1^*(z)$ satisfies the same equation regardless of the value of $z$, it does not depend on $z$ and we only need to solve for the $x$ dependence of this ratio.
>
> &nbsp;
> > _In equations (21)-(23) it is not exactly clear where and why the optimum q1* should be used._
>
> Indeed $q_1^*$ should be used on the *rhs* of Eqs (22-23). We apologize for these typos which will be corrected in the revised manuscript.
>
> &nbsp;
> > _In (22) is the r(x') defined via (14) or via (16) [...]_
>
> Thank you for pointing out this ambiguity. In Eqs (14-15), $q_1$ should be replaced with $q_1^*$. This replacement makes the definitions of $r(x)$ in Eqs (14) and (16) consistent. We will make these edits accordingly in the updated manuscript.
>
> &nbsp;
> > _In Fig. 2 [...] Are these proportionalities derived from or consistent with theory?_
>
> The dashed lines in Fig 2 are not derived from any theory and they serve as a guide to the eye to aid comparisons [Panels (a-c)]. The fact that our predictions in the Gaussian case deviate more from the dashed line guide to the eye than the results of previous approaches does not signify any peculiarity. We will update the figure caption to make this point clear.
>
> &nbsp;
> > _In lines 161-163 you connect the leading corrections for I(Z;Y) and I(Z;X)_
>
> The relations are indeed general and will appear at the end of Sec 3 in our revised manuscript.
>
> &nbsp;
> > _In line 165 you mention that the theory is agnostic about the discrete or continuous nature of the representation [...]_
>
> We apologize for this confusion. Our theory places no restriction on whether the representations are discrete or continuous. However allowing the encoding to take on two different values ($Z=\\{z_0,z_1\\}$) is enough for a complete characterization of the learning onset: we can always choose the unperturbed encoder to be an all-to-one map ($q_0(z_0|x)=1$) and let the linear correction have access to one additional alphabet ($q_1(z_1|x)>0$). We will explain this point in Sec 5 and modify the phrasing in line 184 from "...discrete in this work" to "...can be discrete in our theory". We hope that our answer helps clarify this ambiguity.
>
> &nbsp;
> &nbsp;
>
> __Minor comments__
>
> > _in lines 82-83 [...] the corrections get larger if \epsilon deviates from zero, right?_
>
> Here we actually mean that higher order terms in the power series are smaller because they contains a small parameter $\epsilon$ raised to a higher power. We apologize for the unclear language and will replace the sentence in line 82-83 with *"For small $\epsilon$, the higher order terms in this series represent ever smaller corrections to the unperturbed solution."*
>
> &nbsp;
> > _in line 91, the notation \delta_n,0 is not fully clear. I prefer the notation in eq. (20) or in line 126_
>
> Our revised manuscript will use the notation as suggested.

---

> > ### Comment · Reviewer_o32a · 2021-08-27
> > **Thank You!**
> >
> > Thank you very much for clarifying these issues. I will stick to my score and continue recommending acceptance.

---

### Official Review · Reviewer_DnfR · 2021-07-16

**Rating:** 6
**Confidence:** 3

**Summary:**

This paper investigates the perturbation theory for the information bottleneck method to characterize the learning onset, i.e., the threshold parameter that governs the transition between the informative and uninformative regimes. First order theory and second order perturbation analyses to estimate the learning onset are presented. The utility of the theoretical results is demonstrated with the help of experiments. The authors also show the recovery of known results for Gaussian case using their analysis techniques.

**Main Review:**

1. Since learning onset is central to the theoretical claims in the paper, I expected Section 2 to be more detailed and self contained. It's unclear why the learning onset corresponds to a transition characteristic and why the threshold parameter $\beta_c$ is related to the maximum relevant information encoded per bit.

2. The paper is well organized and mathematical details are reasonably clear, however, some parts of the analysis are not motivated well enough. What's the basis for expansions in (6) and (7)? Are these expansions valid for all stochastic encoders $q(z\mid x)$? What do the quantities $I_{Z;X}^{(1)}[q_1]$ and $I_{Z;X}^{(1)}[q_1,q_2]$ signify?

The paper provides a worthy contribution to the information bottleneck theory by theoretically characterizing the learning onset. However, some parts of the paper could be improved with better motivation and discussions.

**Time Spent Reviewing:**

2

---

> ### Author Response · Authors · 2021-08-10
> **Author Response**
>
> Thank you for your helpful comments and feedback which we will make sure to take into account in our revised manuscript. We respond to your specific questions below.
>
> _1.1 why the learning onset corresponds to a transition characteristic_
>
> The learning onset is a transition because it separates two qualitatively distinct regimes, characterized by $I(Z;Y)=0$ for $\beta<\beta_c$ and $I(Z;Y)>0$ for $\beta>\beta_c$ (see, Fig 1b). In Sec 2 we use the data processing inequality to argue that this transition exists for all $I(X;Y)>0$. We will ensure that this connection is clear in Sec 2 of our updated manuscript.
>
> _1.2 why the threshold parameter $\beta_c$ is related to the maximum relevant information_
>
> To see this relation, we note that the IB frontier, which traces $I(Z;X)$ and $I(Z;Y)$ of the optimal encoders, passes through the origin, is concave down and has a slope equal to $\beta^{-1}$ (Tishby et al, 1999; see, also, Fig 1a). It follows that the relevant information per extracted bit, $I(Z;Y)/I(Z;X)$, is maximum at the origin where the slope is also maximum. In other words this point corresponds to the minimum $\beta$ that admits an informative IB solution. This condition defines the critical parameter $\beta_c$&mdash;hence, the maximum relevant information ratio is equal to $\beta_c^{-1}$ (the maximum slope of the IB frontier).
>
> _2.1 What's the basis for expansions in (6) and (7)?_
>
> The key to developing a perturbation theory is to identify a small controllable parameter of a problem and construct the solution as a power series in that parameter. As this parameter approaches zero, the first few terms of the power series solution provide an accurate approximation of the true solution since higher order terms involve a smaller parameter raised to a higher power. In our work the small parameter corresponds to $\epsilon=\beta-\beta_c$ and we write the IB encoder as a power series in $\epsilon$ in Eq (6). Using Eq (6) to compute $I(Z;X)$ leads directly to Eqs (7-9). We will provide this derivation in Appendix.
>
> _2.2 Are these expansions valid for all stochastic encoders $q(z|x)$?_
>
> In principle this series expansion is valid for all $q(z|x)$ but it is most useful when $\epsilon$ is small in which case the higher order terms become negligible and we can focus only on the first few terms. In our work we consider the limit $\epsilon\to0^+$ as stated below Eq (7).
>
> *2.3 What do the quantities* $I_{Z;X}^{(1)}[q_1]$ *and* $I_{Z;X}^{(2)}[q_1,q_2]$ *signify?*
>
> These quantities capture the first and second-order growths of information as $\beta$ rises above $\beta_c$. In the limit $\beta-\beta_c\to0^+$, the first-order term dominates and the growth of information appears linear (see, Fig 1b). We will provide an explanation of these terms in the updated paper.
>
> __Reference__ \
> N Tishby, FCN Pereira and W Bialek. The information bottleneck method. 37th Allerton Conference on Communication, Control and Computing, 1999

---

### Official Review · Reviewer_hqTY · 2021-07-18

**Rating:** 7
**Confidence:** 4

**Summary:**

The paper studies the learning onset of Information Bottleneck (IB). The authors derive perturbation theory for the IB method and characterize the learning onset, and verifies with experiments. The authors also compare the difference and subtleties in their derivation and previous attempts at deriving a perturbation theory for the learning onset.

**Limitations And Societal Impact:**

The paper is clear about its limitations.

**Main Review:**

In terms of originality, the perturbative approach to address the IB learning onset is not new, but the details of the perturbation, and characterization of the information quantities at the onset is new.

In terms of comparison with the previous work "Learnability for the Information Bottleneck" (Wu et al 2019), there is a difference between the meaning of \epsilon. In (Wu et al 2019), the epsilon is defined as an infinitesimal number which is used for series expansion, which is independent of \beta-\beta_c, and the perturbation on the probability q(z|x) is the global difference between the perturbed probability and the trivial q_0(z|x)=q_0(z), and is allowed on the full support of Z. In the current work, \epsilon is defined as \epsilon=\beta-\beta_c, and therefore there are multiple terms q(z|x) = q0(z|x) + ε q1(z|x) + ε^2 q2(z|x) + ... , w.r.t. \beta-\beta_c. In (Wu et al 2019), the perturbative theory is used for deriving an upper bound of beta_c using sufficient conditions of non-trivial representation, while in the current work, since \epsilon is defined as \epsilon=\beta-\beta_c, it can obtain a tight bound of the onset. In the paper, it will be appropriate if the authors clarify the difference in the above perspective.

Apart from the above, the paper is mostly clear. In terms of significance, the result is incremental, but will be interesting to the IB and representation learning community.

**Time Spent Reviewing:**

1.5 hours

---

> ### Author Response · Authors · 2021-08-10
> **Author Response**
>
> Thank you for your thoughtful comments that will help improve the clarity of our work.
>
> Indeed the set-up of our theory differs slightly from that of Wu et al (2019). In essence we ask how optimal encoders respond to a small change in $\beta$ whereas Wu et al ask how the loss function changes in response to a small perturbation to an encoder. We emphasize however that this difference in perspectives is not the reason why our theory produces a tight bound on the learning onset. Wu et al make an implicit assumption that the representation space is limited to the support of the unperturbed encoder. Importantly this assumption rules out linear corrections to information which are required for the characterization of the learning onset (see Sec 6). We will expand Sec 6 to clarify this point in our revised manuscript.
>
> __Reference__ \
> T Wu, I Fischer, IL Chuang and M Tegmark. Learnability for the information bottleneck. Entropy, 2019

---

### Decision · Program_Chairs · 2021-09-27

**Decision:**

Accept (Poster)

**Comment:**

Reviewers agree that this work is a clear, well-written theory paper that will be of interest to the growing community of researchers working with models trained using the Information Bottleneck and related objective functions. Consequently, we are recommending the paper for acceptance.